# Experimental Design and Bioassays as Tools to Investigate the Impact of Anodic Oxidation on Progestins Degradation

**Juliana Mendonça Silva de Jesus** [1,*], **Allan dos Santos Argolo** [2], **Flávio Kiyoshi Tominaga** [3], **Maria Elena Taqueda** [1], **Daniele Maia Bila** [2], **Sueli Ivone Borrely** [3] **and Antonio Carlos Silva Costa Teixeira** [1,*]

1   Research Group in Advanced Oxidation Processes (AdOx), Department of Chemical Engineering, University of São Paulo, Av. Prof. Luciano Gualberto, 380, São Paulo CEP 05508-010, Brazil
2   Department of Sanitary and Environmental Engineering, Rio de Janeiro State University, Rio de Janeiro CEP 20550-900, Brazil
3   Nuclear and Energy Research Institute (IPEN-CNEN/SP), Av. Prof. Lineu Prestes, 2242, São Paulo CEP 05508-000, Brazil
*   Correspondence: julianams.silva@gmail.com (J.M.S.d.J.); acscteix@usp.br (A.C.S.C.T.)

**Abstract:** The present study investigated the degradation of levonorgestrel (LNG) and gestodene (GES) through an anodic oxidation process mediated by active chlorine species. The independent variables $[LNG]_0$ and $[GES]_0$, current density (mA cm$^{-2}$), and $[NaCl]_0$ (mol L$^{-1}$) were optimized through a response surface methodology (RSM) based on a four-level central composite design (CCD). Specific energy consumption allowed CCD-RSM analysis and optimization. The decay of progestins was followed to verify the kinetics of the anodic degradation process. Chlorine monitoring showed that excess Cl$^-$ concentration did not mean high hormones removal, as well as the excess of current density. Central point conditions ($[NaCl]_0 = 0.07$ mol L$^{-1}$, $j = 32.5$ mA cm$^{-2}$, $[LNG]_0$, and $[GES]_0$ 1.0 mg L$^{-1}$) proved to be the best operational option. The performance with real pharmaceutical wastewater confirmed model optimization (2.2 ± 0.2 kWh g$^{-1}$, with removals of 83.1 ± 0.9% and 75.1 ± 2.8% for LNG and GES, respectively). The selected condition was used for estrogenic activity and acute toxicity assays. The first allowed the identification of the initial estrogenic activity for the mixture of LNG and GES (924 E2-EQ ng L$^{-1}$). Additionally, the electrochemical process could decrease this environmental parameter by 74.6%. The progestin mixture was classified as acute toxicity to *Daphnia similis*, with a toxicity unit (TU) of 2.5 100/EC50%. After electrolysis, the hormone solutions reached a fourfold increase in TU value, classified as high acute toxicity.

**Keywords:** central composite design; progestins; real pharmaceutical wastewater; estrogenic activity; acute toxicity

## 1. Introduction

The fate and occurrence of endocrine disrupting compounds (EDCs) promoted an emerging concern worldwide. As a result of anthropogenic actions, EDCs have been found in distinct environmental matrices, such as water, soil, and air. Among them, synthetic hormones, such as progestins, have been used in ecotoxicological assays to highlight their hazard to *Dreissena polymorpha* [1], fathead minnow [2], and zebrafish [3].

Levonorgestrel (LNG) and gestodene (GES) are progestins used in hormonal maintenance via contraceptive pills and intrauterine devices, whose consumption has been increasing yearly. According to the United Nations [4], 151 million women worldwide chose the contraceptive pill as their primary contraceptive method; in Brazil, 2.6 million emergency contraceptive pills (ECPs) were consumed in 2017 [5]. ECPs have LNG as their only active principle (1.50 mg), besides using GES and LNG in a combined formulation with estrogens, in doses of 0.075 to 0.15 mg, respectively [6].

Pharmaceutical production and consumption levels are related to the number of substances discharged into environmental matrices (Table 1). The primary sources of

micropollutants are household, pharmaceutical, and livestock farm runoff, which flow out into wastewater treatment plants (WWTP) [7–10]. Administered hormones are not fully metabolized by humans and animals. In human metabolism, about 77% and 10% of ingested LNG and GES are eliminated in their active form via urine, respectively [8,9].

**Table 1.** Concentration of progestins in water matrices (ng $L^{-1}$).

| Matrices | Progestin | | References |
| --- | --- | --- | --- |
| | **Levonorgestrel (LNG)** | **Gestodene (GES)** | |
| Surface water | 5.3–7.0 | 5.03 | [8,9,11] |
| Wastewater treatment plants (WWTPs) | <0.2–16.1 (influent) <0.5–4.0 (effluent) | 0.44 (influent) 0.19 (effluent) | [8,11–13] |
| Pharmaceutical wastewater | $4$–$5 \times 10^6$ (total concentration) 0.8–$0.16 \times 10^5$ (aqueous concentration) | $0.66 \times 10^5$ | [14,15] |

In the last 26 years, studies have investigated electrochemical advanced oxidation processes (EAOPs) to remove hormones from water matrices, among which hormones have been used as the target pollutant. A brief search in the Scopus and Web of Science online platforms using the keywords "anodic oxidation" AND "hormones" showed 24 and 28 articles. In this sense, Table 2 presents the main studies evaluating the effectiveness of anodic oxidation for the degradation of hormones in synthetic and real matrices.

**Table 2.** Anodic oxidation (AO) studies applied for the removal of hormones.

| Hormone | Concentration | Matrices | Electrodes | Support Electrolyte | Current Density | Reference |
| --- | --- | --- | --- | --- | --- | --- |
| Progesterone | 0.1 to 100 mg $L^{-1}$ | Milli-Q water | BDD (anode)/SS (cathode) | 0.035 mol $L^{-1}$ $Na_2SO_4$ or NaCl | 15–100 mA | [16] |
| Estrone | 230, 410, and, 570 µg $L^{-1}$ | Milli-Q water | BDD (anode)/SS (cathode) | 0.36 mol $L^{-1}$ NaCl | 5, 10, and, 25 mA cm$^{-2}$ | [17] |
| Estradiol (E2) and Ethyniletradiol (EE2) | 1 mg $L^{-1}$ | Milli-Q water and WWTP | BDD (anode)/zirconium (cathode) | 0.1 mol $L^{-1}$ Na2SO4 or NaCl | 0.9–2.6 mA cm$^{-2}$ | [18] |
| Gestodene (GES) and Ethyniletradiol (EE2) | 625 µg $L^{-1}$ of GES and 250 µg $L^{-1}$ of EE2 | Milli-Q water | BDD as anode and cathode | 0.02, 0.05 and 0.1 mol $L^{-1}$ $Na_2SO_4$ | 12, 32 and, 48 mA cm$^{-2}$ | [19] |
| 17-α-Methyltestosterone | 5.0 mg $L^{-1}$ | Milli-Q water | DSA as anode and cathode | 0.5 mol $L^{-1}$ Na2SO4 or NaCl | 15, 30, and 45 mA cm$^{-2}$ | [20] |
| Ethyniletradiol (EE2) | 100 mg $L^{-1}$ | Water/methanol 7: 3 *v/v* | DSA (anode)/ SS (cathode) | 0.004 mol $L^{-1}$ NaCl | 40 mA cm$^{-2}$ | [21] |
| Estradiol (E2) | 250–750 µg dm$^{-3}$ | Water | BDD-WE Hg/Hg$_2$Cl$_2$·KCl (sat.) (SCE)-RE Pt-CE | 0.1 mol $L^{-1}$ $Na_2SO_4$, NaNO3, and NaCl | 25 mA cm$^{-2}$ | [22] |
| Levonorgestrel (LNG) | 2.5 mg $L^{-1}$ | Milli-Q water and pharmaceutical wastewater | PVC-graphite (anode)/platinum (cathode) | 0.05, 0.1, and 0.2 g NaCl | 3, 6, and 8 V | [23] |

SS = stainless steel, WE = working electrode, RE = reference electrode, CE = counter electrode.

Based on that, the electrodes applied to the degradation of organic compounds are classified as either active or non-active. The first is characterized by the strong affinity of HO$^\bullet$ radicals (the primary oxidizing species in AOPs) with the surface of the electrode (M), reducing the number of radicals that will be available to directly react with the target contaminant (Equations (1) and (2)). This behavior is observed for $RuO_2$, $IrO_2$, and Pt, for example [24–26].

$$H_2O + M \rightarrow M(^\bullet OH) + H^+ + e^- \tag{1}$$

$$M(^\bullet OH) \rightarrow M + \frac{1}{2} O_2 + H^+ + e^- \tag{2}$$

In contrast, electrodes such as $SnO_2$ and boron-doped diamond (BDD) can easily generate and release $^\bullet$OH radicals to the solution, improving degradation and potentially leading to the total mineralization of the contaminants ($R_{(aq)}$) [24,26–28]:

$$R_{(aq)} + M(^\bullet OH)_{n/2} \rightarrow M + \text{transformation products} + n/2\ H^+ + n/2\ e^- \tag{3}$$

In addition to hydroxyl radicals, EAOPs can mediate the generation of other strong oxidants, such as chlorine ($Cl^-$), persulfate ($S_2O_8^{2-}$), or superphosphate ($P_2O_8^{4-}$). This occurs in parallel to the electron generation of HO$^\bullet$ [24–26].

However, to obtain the maximum performance of contaminant removal by EAOP, it is necessary to identify the main factors that affect these processes, such as the initial concentration of the target contaminant(s), the concentration of oxidants, and the current density. All these parameters are modified depending on the process studied and the matrices [29]. Furthermore, the best operating condition must achieve not only high removal rates but also low energy consumption and little generation of toxic by-products. To this end, EAOP studies are usually performed together with ecotoxicological, cytotoxic, and estrogenic activity tests. Such experimental proposals contribute to motivating the use of EAOPs, and under some conditions, the combined oxidative process reaches the potential of pollutant mineralization and toxicity removal [30–32].

Bioassays have been used to confirm the effectiveness of removing hazardous chemicals from target matrices [33–35]. For example, the in vitro yeast estrogen screen (YES) assay has been applied to environmental samples. The advantages of YES include simple, easy handling; low costs; non-invasive, non-sample pre-treatment; and the possibility of dealing with trace concentration levels [36,37]. YES is based on the yeast *Saccharomyces cerevisiae* modified with a human estrogen receptor, which quantifies the total estrogenic potential of samples, considering the synergistic effects of multiple substances [37]. In complex environmental matrices, the matrix effect can inhibit the estrogen receptor, and, consequently, the response. However, cytotoxicity (yeast growth inhibition) and the presence of antagonistic substances are other pieces of evidence obtained from the assay. Estrogenic activity can still be determined [36–38].

In turn, ecotoxicity assays have been used to improve the risk assessment network on emerging contaminants, including hormones. Standard procedures and test organisms have been applied, e.g., *Daphnia similis* [39], *Daphnia magna* [40], and *Vibrio fischeri* [41]. Evaluation of the toxicity of organic compounds in aqueous matrices before and after treatment is recommended to motivate the inclusion of such compounds in current environmental legislation and highlight viable possibilities for ecological treatment and monitoring [41–43].

In this context, the present work aimed to optimize and determine the experimental conditions that allow low specific energy consumption (SEC) and removals ≥70% of both progestogens in synthetic solutions and real pharmaceutical wastewater (RPW). To this end, the experiments were conducted following a central composite experimental design coupled with response surface analysis. The study of hormones concentration as an independent variable, in the mixture, is a novelty among the various studies that address the optimization of advanced electrochemical processes [35,44,45], considering the future application of the process in an industrial environment where the concentration

of such compounds is not fixed a priori. From this, it was possible to determine a work region in which lower energy consumption and high removals of both hormones can be obtained, close to the central conditions of the experimental domain. Additionally, the study brings the ecotoxicological verification of such conditions, which if applied, can provide a reduction in the initial estrogenic activity. In brief, the main objectives were: (i) to investigate LNG and GES degradation in a synthetic mixture and real pharmaceutical wastewater, through anodic oxidation; (ii) to evaluate the effects of initial progestin concentrations, current density, and NaCl concentration through a central composite experimental design; (iii) evaluate the anodic process through acute toxicity tests with *Daphnia similis*; and (iv) to assess the estrogenicity of progestin samples, before and after the anodic process, using the yeast estrogen screen (YES) assay.

## 2. Materials and Methods

### 2.1. Reagents

LNG ($C_{21}H_{28}O_2$, $\geq 98.0\%$, CAS 797-63-7) and GES ($C_{21}H_{26}O_2$, $\geq 98.0$, CAS 60282-87-3) were purchased from Zhejiang Xianju Pharmaceutical Co. Ltd. and used as analytical standards and in the preparation of working solutions. In addition, 17 β-estradiol (E2, $C_{18}H_{24}O_2$, $\geq 98\%$, CAS 50-28-2) was purchased from Sigma-Aldrich. Chlorophenol red-β-D-galactopyranoside (CPRG), sodium chloride (NaCl, $\geq 99.0\%$), and sodium thiosulfate pentahydrate ($Na_2S_2O_3 \cdot 5\ H_2O$, $\geq 99.0\%$) were purchased from Merck, Casa Americana Ltd., and Vetec Ltd., respectively. HPLC grade methanol ($CH_3OH$), ethanol ($C_2H_5OH$), and acetic acid ($CH_3CO_2H$) were purchased from Sigma-Aldrich. All solutions were prepared using ultrapure water (18.2 MΩ cm) from a Milli-Q® system (Merck Millipore, Burlington, MA, USA).

### 2.2. Electrochemical Degradation Studies

An undivided electrochemical cell was used, containing DSA® ($Ti/Ru_{0.3}Ti_{0.7}O_2$, De Nora do Brasil Ltd., Brazil) and stainless-steel electrodes as the anode and cathode, respectively. With 28 cm$^{-2}$ of the active area, both electrodes were parallel at 13 mm. The power supply (30V/5A MPS 300-5B MINIPA) was used as a constant current source. NaCl was used as the electrolyte. The solutions (300 mL) were recirculated from a borossilicate glassreservoir to the electrochemical cell with the aid of a peristaltic pump (DM 500) in a continuous flow of 0.75 mL s$^{-1}$ (Figure S1). The internal volume of the electrochemical cell was 36.4 mL, and all experiments were performed at a space-time of 16 s in each pass. The conditions of the experiments (NaCl and progestin concentrations; current density) are summarized in Table 3. Progestin solutions were sampled at 0, 2, 4, 6, 8, 10, 15, and 20 min for analysis. The sample volume was 900 L, to which 100 μL of $Na_2S_2O_3$ solution (10 g L$^{-1}$) was added to stop any further reaction [46].

**Table 3.** Independent variables and their coded and uncoded values.

| Independent Variables | Symbol | Coded Levels | | | | |
| --- | --- | --- | --- | --- | --- | --- |
| | | −2 | −1 | 0 | +1 | +2 |
| | | Uncoded Values | | | | |
| [GES]$_0$ (mg L$^{-1}$) | $X_1$ | 0.0 | 0.5 | 1.0 | 1.5 | 2.0 |
| [LNG]$_0$ (mg L$^{-1}$) | $X_2$ | 0.0 | 0.5 | 1.0 | 1.5 | 2.0 |
| $j$ (mA cm$^{-2}$) | $X_3$ | 7.5 | 20.0 | 32.5 | 45.0 | 57.5 |
| [NaCl]$_0$ (mol L$^{-1}$) | $X_4$ | 0.01 | 0.04 | 0.07 | 0.10 | 0.13 |

### 2.3. Analytical Methods

Ultra-fast liquid chromatography (UFLC) was performed using a Shimadzu (SPD 10 AV) device equipped with a UV-visible detector (SPD 20A, Shimadzu, Kyoto, Japan) and a C18 column (ACE, 250 mm × 4.60 mm, 5 μm, ThermoFisher Scientific, Waltham, MA, USA), for the simultaneous monitoring of GES and LNG concentrations. An isocratic

method was applied, with 70% of MeOH and 30% ultrapure water containing 1% $v/v$ of acetic acid as the mobile phase.

LNG and GES were detected at 244 nm. The sample injection volume, oven temperature, and flow rate were 20 μL, 40 °C, and 0.2 mL min$^{-1}$, respectively. Under these conditions, the retention times of LNG and GES were 10.0 and 8.0 min, respectively. Due to the low water solubility of LNG (Table S1), stock solutions of LNG and GES (10 mg L$^{-1}$) were prepared in methanol and used to spike progestin standards in concentrations of 0.05–10.0 mg L$^{-1}$.

A calibration curve was obtained, and validation parameters were determined, where LOD and LOQ refer to detection and quantification limits, respectively. For LNG: ($R^2 = 0.9991$, LOD = 20 μg L$^{-1}$ and LOQ = 70 μg L$^{-1}$); for GES: ($R^2 = 0.9980$, LOD = 60 μg L$^{-1}$, LOQ = 200 μg L$^{-1}$). The efficiency of progestin removal was calculated by Equation (4):

$$progestin(\%) = \frac{(C_0 - C_t)}{C_0} * 100 \tag{4}$$

where $C_0$ and $C_t$ refer to initial and final LNG and GES concentrations (mg L$^{-1}$), respectively.

### 2.4. Central Composite Design (CCD)

Considering the standard order of the CCD approach, 31 experiments were conducted based on seven replications of the central point, eight-star points, and 18 factorial points in a random sequence. Minitab 20 was used for regression and analysis of the data obtained. ANOVA allowed the evaluation of polynomial response surface models (RSM) at a 95% confidence interval. In this investigation, parameters such as F-value, $p$-value, and $R^2$ were applied for data confirmation and accuracy. The F-value was the ratio of the parameter's variance to the error variance (residual variance) or the ratio of the mean square parameter to the mean square error. RSM was applied for identifying the optimum operating conditions for the degradation of LNG and GES, aiming to minimize energy consumption.

CCD was applied for evaluating the effect of four independent variables: initial NaCl concentration (mol L$^{-1}$), current density $j$ (mA cm$^2$), and initial LNG and GES concentrations (mg L$^{-1}$). Table 3 presents these parameters, converted to dimensionless values coded as $X_1$, $X_2$, $X_3$, and $X_4$ at five levels ($-2$, $-1$, $0$, $+1$, and $+2$). The variable levels were selected based on the progestins' solubility (Table S1) and previous studies [47].

A second-order polynomial multiple regression equation was applied to represent the relationship between the response (SEC, specific energy consumption) and the operational parameters. The corresponding quadratic equation, along with the interaction effects of independent variables, is depicted in Equation (5):

$$SEC \left(\text{kWh g}^{-1}\right) = \beta_0 + \sum_{i=1}^{n} \beta_i X_i + \sum_{i=1}^{n} \beta_{ii} X_i^2 + \sum_{i=1}^{n-1} \sum_{j=i+1}^{n} \beta_{ij} X_i X_j + \varepsilon \tag{5}$$

where SEC is the predicted response variable; $\beta_0$ is the intercept; $\beta_i$, $\beta_{ij}$, and $\beta_{ii}$ are the coefficients of the linear effect and double interactions; and $X_i$ and $X_j$ refer to the independent variables or factors. The specific energy consumption (SEC) is defined as the amount of energy consumed in kWh to degrade 1 g of organic load from an effluent—in this case, progestins, according to Equation (6):

$$SEC \left(\text{kWh g}^{-1}\right) = \left(\frac{\text{UIt}}{\Delta[\text{progestin}]\text{V}}\right) \tag{6}$$

where U is the average potential of the electrochemical cell (V); t is the electrolysis time (h); I is the electric current (A); $\Delta$[progestin] corresponds to the variation in LNG and GES concentrations (g L$^{-1}$); and V is the volume of the working solution (L).

*2.5. In Vitro Estrogenic Assay—YES*

The estrogenicity of progestin samples was assessed before and after anodic oxidation with YES, an in vitro recombinant reporter gene assay [48]. Before the YES assay, progestin samples were concentrated through C18 cartridges (SPE Strata 200 mg/3 mL).

The SPE protocol consisted of the following steps: (i) cartridge conditioning with 10 mL of methanol and 10 mL of pure water; (ii) percolation of progestin samples at 4 mL min$^{-1}$; (iii) rinse with 10 mL of methanol (2% $v/v$); and (iv) elution of LNG and GES analytes with 2 mL of ethanol. LNG and GES extraction recoveries were (91.60 ± 0.04) % and (84.90 ± 2.47) %, respectively.

The bioassay was performed in 96-well microplates with serial dilution of sample extracts in ethanol. E2 was used as the positive control and standard curve, 2724 to 1.33 ng L$^{-1}$, and ethanol was used as the negative control. An amount of 10 μL of each sample dilution was transferred to a test plate and allowed to evaporate. Then, 200 μL of culture medium containing yeast and the chromogenic substrate chlorophenol red-β-D-galactopyranoside (CPRG) were added. After incubation for 72 h at 30 °C, absorbances at 575 nm and 620 nm were measured with a VersaMax microplate reader (Molecular Devices, San Jose, CA, USA).

Data Analysis

Measured absorbances were corrected with Equation (7) to discount the turbidity effect from the estrogenic response:

$$Abs_{corr\ (sample)} = Abs_{575\ (sample)} - \left(Abs_{620\ (sample)} - Abs_{620\ (blanks)}\right) \tag{7}$$

Absorption and corrected concentration data were plotted, and the resulting sigmoidal curves were fitted to a symmetric logistic function using the software Origin 2020 (Origin-Lab). The estradiol equivalent (E2-EQ) results were obtained in ng L$^{-1}$ by interpolating the standard E2 dose-response curve (Figure S3) and the sample data with a log-logistic model:

$$y = \frac{A_1 - A_2}{1 + (x_0/x)^p} + A_2 \tag{8}$$

where $A_1$ and $A_2$ refer to the maximum and minimum β-galactosidase induction in corrected absorbance, $x_0$ is the median effect concentration EC50% for E2 in ng L$^{-1}$, $p$ corresponds to the slope of the sigmoidal curve, and $(x, y)$ is the ordered pair related to a sample concentration and its response in corrected absorbance. Finally, the E2-EQ was calculated as the lowest $x$ that elucidated an agonist response divided by the final sample enrichment factor in the assay [38].

*2.6. Ecotoxicity Assays*

Acute toxic assays were performed with the microcrustacean *D. similis* according to the Brazilian standard method, ABNT NBR 12713/2016 [49]. Daphnids were cultivated at the Laboratory of Biological and Environmental Assays (Nuclear and Energy Research Institute-IPEN-CNEN/SP, São Paulo, Brazil). The assays evaluated the effects of progestins in the mixture on the test organisms, before and after anodic oxidation conducted under central point conditions (Table 3). The immobility of *D. similis* after 48 h was the endpoint measured for this assay. For this purpose, neonates between 6 h and 24 h of age were exposed to several dilutions of treated and non-treated progestin solutions for 48 h.

The following solutions were evaluated through acute toxic assays: (A) a mixture of progestogens (LNG + GES) without electrochemical processing and in the absence of NaCl; (B) a mixture of progestogens (LNG + GES) without electrochemical processing in the presence of NaCl; (C) a NaCl solution without electrochemical processing; (D) a NaCl solution after electrochemical processing; and (E) a mixture of progestogens (LNG + GES) after electrochemical processing in the presence of NaCl. All samples had pH

values corrected to 7.0 before toxicity assays, as requested by ABNT NBR 12713/2016 [49]. Acute toxicity is expressed in toxicity units (TU = 100/EC50%) and corresponds to the average effect concentrations that promoted 50% immobility of exposed living organisms (EC50%-48 h, expressed in % *v/v*).

## 3. Results and Discussion

### 3.1. Hormones Degradation and Response Surface Modeling

Table 4 presents the 31 experiments carried out in the standard CCD order, together with the values of SEC, percentage removals, and specific removal rates obtained for LNG and GES after 6 min of electrochemical treatment, with their respective coefficients of determination. Among the experimental conditions considered to evaluate the effectiveness of the anodic oxidation in the joint removal of progestins, runs 19 and 17 brought the removal and determination of the specific energy consumption for the treatment of solutions containing only LNG or GES, respectively. As a result, the lowest SEC of the entire experimental design (0.4 kWh $g^{-1}$) was obtained in experiment 19, when the electrochemical system was applied to treat the solution containing 1.0 mg $L^{-1}$ of GES with $[NaCl]_0$ = 0.07 mol $L^{-1}$ and current density of 32.5 mA $cm^{-2}$. In addition to the minimum SEC, 96.4% GES was removed, with a specific degradation rate equal to 0.493 $min^{-1}$. On the other hand, in run 17, performed only with LNG under the same experimental conditions, an SEC value eleven times greater than that observed for the GES solution was obtained. Furthermore, the percent removal and $k_{obs}$ for LNG were 78.1% and 0.212 $min^{-1}$, respectively; thus, these results indicate different degradation behaviors exhibited by the hormones in relation to the anodic electrochemical process.

A closer inspection of the results in Table 4 shows that for experiments 1 and 5, in which the progestogen concentrations were 0.5 mg $L^{-1}$ and the NaCl concentration was 0.04 mol $L^{-1}$, it is observed that the effect of variable $X_3$ (*j*) on hormone degradation resulted in higher LNG and GES removals (84.2% and 100%, respectively) for the lowest current density, 20 mA $cm^{-2}$. On the other hand, for experiments 4 and 8, which started with concentrations of progestins three times higher (1.5 mg $L^{-1}$) and the same concentration of electrolyte, the opposite effect was observed; that is, for higher initial concentrations of hormone it was necessary to use high values of current density (*j* = 45 mA $cm^{-2}$) to achieve greater removal of hormones. The same is observed when comparing experiments 9 and 13, conducted with the lowest hormone concentrations, however, with a concentration of electrolyte 2.5 times higher. These results suggest that: (i) For initial concentrations of progestins $\leq$ 1.0 mg $L^{-1}$, the current density has a positive impact on the removal; however, when exceeding such concentration, this density is insufficient. That is, the process is being operated above the threshold current density. Periyasamy et al. [50] report that increasing the current density increases the voltage, favoring the evolution of oxygen and disfavoring the evolution of chlorine. In this case, the oxygen generated in the electrode results in a reduction in the efficiency of the removal of organic molecules, inhibiting the transfer of mass to the electrode surface. (ii) There is a strong interaction between the variable's current density and electrolyte concentration, as well as between the current density and the concentration of progestogens (in particular gestodene) so that the effect of increasing the current density will markedly depend on the levels of $[NaCl]_0$ and $[GES]_0$. In fact, for intermediate concentrations of LNG and GES, electrolyte content, and *j* = 32.5 mA $cm^{-2}$ (central point), satisfactory SEC values (1.9 $\pm$ 0.2 kWh $g^{-1}$) and progestin removals greater than 70%, with good specific removal rates (0.270 $\pm$ 0.1 $min^{-1}$ and 0.242 $\pm$ 0.1 $min^{-1}$ for LNG and GES, respectively) were obtained, in line with the study objectives.

**Table 4.** CCD matrix and variable levels used to study the influence of operating parameters on the anodic oxidation of levonorgestrel (LNG) and gestodene (GES), response variable (SEC) applied in the statistical analysis, and additional observed results.

| Run | Coded Levels | | | | Experimental Variable Levels | | | | Response | Removal Efficiency (%) | | $k_{obs}$ (min$^{-1}$) | | | |
|---|---|---|---|---|---|---|---|---|---|---|---|---|---|---|---|
| | $X_1$ | $X_2$ | $X_3$ | $X_4$ | $[GES]_0$ (mg L$^{-1}$) | $[LNG]_0$ (mg L$^{-1}$) | $j$ (mA cm$^{-2}$) | $[NaCl]$ (mol L$^{-1}$) | SEC (kWh g$^{-1}$) | LNG | GES | LNG | $R^2$ | GES | $R^2$ |
| 1 | −1 | −1 | −1 | −1 | 0.5 | 0.5 | 20 | 0.04 | 1.3 | 84.2 | 100.0 | 0.2673 | 0.9427 | 0.7374 | 0.8403 |
| 2 | +1 | −1 | −1 | −1 | 1.5 | 0.5 | 20 | 0.04 | 2.1 | 47.6 | 86.4 | 0.0874 | 0.9122 | 0.2856 | 0.9360 |
| 3 | −1 | +1 | −1 | −1 | 0.5 | 1.5 | 20 | 0.04 | 10.1 | 41.1 | 80.2 | 0.0761 | 0.8989 | 0.2717 | 0.9999 |
| 4 | +1 | +1 | −1 | −1 | 1.5 | 1.5 | 20 | 0.04 | 6.4 | 35.2 | 34.3 | 0.0746 | 0.9934 | 0.0672 | 0.9965 |
| 5 | −1 | −1 | +1 | −1 | 0.5 | 0.5 | 45 | 0.04 | 1.8 | 46.9 | 79.9 | 0.1242 | 0.9442 | 0.2421 | 0.9796 |
| 6 | +1 | −1 | +1 | −1 | 1.5 | 0.5 | 45 | 0.04 | 0.9 | 56.1 | 81.1 | 0.1219 | 0.9703 | 0.2992 | 0.9273 |
| 7 | −1 | +1 | +1 | −1 | 0.5 | 1.5 | 45 | 0.04 | 4.4 | 73.5 | 100.0 | 0.2192 | 0.9997 | 0.1798 | 0.9921 |
| 8 | +1 | +1 | +1 | −1 | 1.5 | 1.5 | 45 | 0.04 | 6.2 | 83.6 | 36.7 | 0.1850 | 0.9388 | 0.0594 | 0.8617 |
| 9 | −1 | −1 | −1 | +1 | 0.5 | 0.5 | 20 | 0.10 | 0.9 | 58.5 | 50.3 | 0.2688 | 0.9717 | 0.0964 | 0.9032 |
| 10 | +1 | −1 | −1 | +1 | 1.5 | 0.5 | 20 | 0.10 | 0.9 | 60.6 | 76.1 | 0.1757 | 0.8838 | 0.1993 | 0.9349 |
| 11 | −1 | +1 | −1 | +1 | 0.5 | 1.5 | 20 | 0.10 | 3.8 | 79.8 | 96.6 | 0.3204 | 0.9600 | 0.5381 | 0.9692 |
| 12 | +1 | +1 | −1 | +1 | 1.5 | 1.5 | 20 | 0.10 | 2.9 | 47.5 | 39.5 | 0.1209 | 0.9772 | 0.0719 | 0.9429 |
| 13 | −1 | −1 | +1 | +1 | 0.5 | 0.5 | 45 | 0.10 | 2.2 | 79.2 | 69.6 | 0.2180 | 0.9247 | 0.1776 | 0.9650 |
| 14 | +1 | −1 | +1 | +1 | 1.5 | 0.5 | 45 | 0.10 | 1.0 | 55.3 | 91.5 | 0.1447 | 0.9896 | 0.4209 | 0.9660 |
| 15 | −1 | +1 | +1 | +1 | 0.5 | 1.5 | 45 | 0.10 | 5.1 | 29.0 | 90.8 | 0.0548 | 0.9815 | 0.3586 | 0.9691 |
| 16 | +1 | +1 | +1 | +1 | 1.5 | 1.5 | 45 | 0.10 | 2.7 | 54.4 | 55.3 | 0.1100 | 0.9322 | 0.1616 | 0.9373 |
| 17 | −2 | 0 | 0 | 0 | 0.0 | 1.0 | 32.5 | 0.07 | 4.5 | 78.1 | - | 0.2124 | 0.9389 | 0.0000 | 0.0000 |
| 18 | +2 | 0 | 0 | 0 | 2.0 | 1.0 | 32.5 | 0.07 | 2.0 | 75.0 | 50.3 | 0.2681 | 0.9671 | 0.0940 | 0.9053 |
| 19 | 0 | −2 | 0 | 0 | 1.0 | 0.0 | 32.5 | 0.07 | 0.4 | - | 96.4 | 0.0000 | 0.0000 | 0.4931 | 0.9366 |
| 20 | 0 | +2 | 0 | 0 | 1.0 | 2.0 | 32.5 | 0.07 | 7.0 | 45.4 | 87.7 | 0.1017 | 0.9962 | 0.3325 | 0.9830 |
| 21 | 0 | 0 | −2 | 0 | 1.0 | 1.0 | 7.5 | 0.07 | 5.8 | 39.4 | 42.5 | 0.0806 | 0.9975 | 0.0891 | 0.9981 |
| 22 | 0 | 0 | +2 | 0 | 1.0 | 1.0 | 57.5 | 0.07 | 1.8 | 37.7 | 89.5 | 0.0771 | 0.9986 | 0.4221 | 0.9975 |
| 23 | 0 | 0 | 0 | −2 | 1.0 | 1.0 | 32.5 | 0.01 | 7.5 | 51.7 | 66.6 | 0.1041 | 0.7694 | 0.1297 | 0.7667 |
| 24 | 0 | 0 | 0 | +2 | 1.0 | 1.0 | 32.5 | 0.13 | 2.7 | 42.6 | 49.9 | 0.0891 | 0.9974 | 0.1134 | 0.9981 |
| 25 | 0 | 0 | 0 | 0 | 1.0 | 1.0 | 32.5 | 0.07 | 2.0 | 90.4 | 83.6 | 0.3456 | 0.9456 | 0.2862 | 0.8021 |
| 26 | 0 | 0 | 0 | 0 | 1.0 | 1.0 | 32.5 | 0.07 | 1.8 | 87.1 | 73.2 | 0.2516 | 0.8125 | 0.2796 | 0.8570 |
| 27 | 0 | 0 | 0 | 0 | 1.0 | 1.0 | 32.5 | 0.07 | 2.1 | 86.2 | 87.5 | 0.3454 | 0.9256 | 0.3608 | 0.9870 |
| 28 | 0 | 0 | 0 | 0 | 1.0 | 1.0 | 32.5 | 0.07 | 1.6 | 68.6 | 63.0 | 0.2566 | 0.9477 | 0.1344 | 0.9000 |
| 29 | 0 | 0 | 0 | 0 | 1.0 | 1.0 | 32.5 | 0.07 | 2.2 | 81.0 | 67.6 | 0.2244 | 0.9092 | 0.1899 | 0.9950 |
| 30 | 0 | 0 | 0 | 0 | 1.0 | 1.0 | 32.5 | 0.07 | 1.7 | 80.4 | 72.9 | 0.2250 | 0.9270 | 0.2160 | 0.9977 |
| 31 | 0 | 0 | 0 | 0 | 1.0 | 1.0 | 32.5 | 0.07 | 2.0 | 79.5 | 76.4 | 0.2422 | 0.9849 | 0.2250 | 0.9916 |

The CCD-RSM was applied as a tool to identify and optimize the effects of the four main operating variables on the anodic oxidation of LNG and GES in a mixture using the DSA-Cl$_2$ system to determine the experimental region that provides the lowest SEC and progestin removals ≥70%. Thus, a quadratic equation (Equation (9)), describing the mutual relationships between the experimental parameters and the response (SEC) was obtained from the significant effects (coefficients with $p < 0.001$), also observed through the Pareto chart (Figure S2).

$$\text{SEC}\left(\text{kWh g}^{-1}\right) = 2.1916 - 0.4777X_1 + 1.8254X_2 - 0.5044X_3 - 0.9801X_4 + 0.1399X_1^2 + 0.2701X_2^2 + 0.3016X_3^2$$
$$+0.6106X_4^2 - 0.2407X_1X_2 - 0.1635X_1X_4 - 0.3329X_2X_3 - 0.7170X_2X_4 + 0.5627X_3X_4 \tag{9}$$

where $X_1$, $X_2$, $X_3$, and $X_4$ are the coded experimental parameters; $(X_1)(X_2)$, $(X_1)(X_4)$, $(X_2)(X_3)$, $(X_2)(X_4)$, $(X_3)(X_4)$ correspond to the interactions of quadratic terms; and $(X_1)^2$, $(X_2)^2$, $(X_3)^2$ and $(X_4)^2$ are the second-order terms. The ranges and levels of the experimental parameters are shown in Table 3. The predicted specific energy consumption (SEC) observed in progestin removal versus the experimental data is shown in Table 4. Furthermore, the ANOVA was implemented to determine the significance and adequacy of the statistical method (Table 5).

**Table 5.** Analysis of variance (ANOVA) for the quadratic model and F-test.

| Source | Sum of Squares | Degree of Freedom | Mean Square | F-Value | *p*-Value |
|---|---|---|---|---|---|
| Regression | 146.99 | 14 | 10.50 | 188.493 | 0.004 [a] |
| First-order effects | 114.60 | 4 | 28.65 | 2057.37 | 0.0001 [a] |
| Second-order effects | 15.90 | 4 | 3.97 | 285.60 | 0.0207 [a] |
| Residual | 16.93 | 16 | 1.05 | | |
| Interactions | | | | | |
| $[GES]_0 \times [LNG]_0$ | 0.92 | 1 | 0.92 | 16.63 | 0.006 [a] |
| $[GES]_0 \times [NaCl]_0$ | 0.42 | 1 | 0.42 | 7.674 | 0.032 [a] |
| $[LNG]_0 \times j$ | 1.77 | 1 | 1.77 | 31.82 | 0.0013 [a] |
| $[LNG]_0 \times [NaCl]_0$ | 8.22 | 1 | 8.22 | 147.65 | 0.00001 [a] |
| Lack of fit | 16.59 | 10 | 1.66 | 29.78 | 0.0002 [a] |
| Pure error | 0.33 | 6 | 0.05 | | |
| Total | 163.92 | 30 | | | |

$R^2 = 0.8951$; [a] Significance at 95% confidence level.

The main observation on the model equation (Equation (9)) and the Pareto chart (Figure S2) is associated with the variable $X_2$ ($[LNG]_0$), given the significance of first-order effects, second-order effects, and interactions. In the first two cases, the effects were positive; that is, the increase in the concentration of LNG contributes to the increase in energy consumption. On the other hand, Figure S2 highlights the non-significant interaction effect of variables $X_1$ ($[GES]_0$) and $X_3$ (*j*), with no impact on specific energy consumption, which was not included in Equation (9).

The desirable value for "adequate precision" is F-value > 4.0. As shown in Table 5, the value obtained for "adequate precision" was 188.5. In addition, the fitted regression model was generally significant with a *p*-value <0.001, which was lower than 0.004, and the lack of fit (*p*-value = 0.004) was insignificant, indicating the suitability of the statistical model.

The evaluation of the regression model was also performed using Figure 1. As can be seen, Figure 1a indicates a strong correlation between model predictions and experimental values. Regarding the raw residuals (Figure 1b), the symmetrical distribution of the data and the clustering tendency in the middle of the plot is further evidence to confirm the quality of the model.

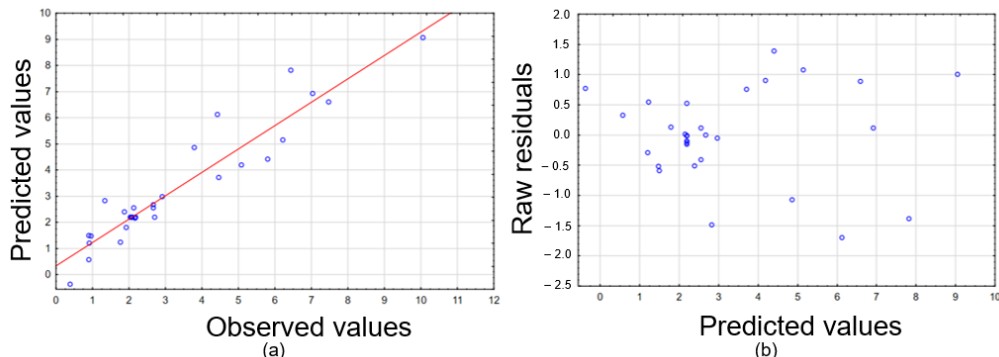

**Figure 1.** Predicted vs. observed values (**a**) and (**b**) raw residuals vs. predicted values for the specific energy consumption (SEC kWh g$^{-1}$).

### 3.1.1. Effect of Operating Parameters

The CCD-RSM approach allows for the possibility of examining and interpreting both the underlying effects and their interactions. This interpretation was made using nonlinear forms in a multidimensional space. It is possible to interpret the effects of parameters on steady levels of the target variable using the response surface procedure (Figure 2).

As shown in Figure 2a, at the central levels of $j$ = 32.5 mA cm$^2$ ($X_3$) and $[NaCl]_0$ = 0.07 mol L$^{-1}$($X_4$), the average response (SEC) was estimated to increase for high levels of $X_3$ and low levels of $X_4$. This observation was also verified using the model (Equation (9)), in which the coefficient of the interaction between these variables had a positive effect on the response (+ 0.5627$X_3X_4$) with a $p$-value < 0.001, indicating significant interaction.

The anodic oxidation in the presence of chloride anions leads to the formation of chlorine, hypochlorous acid, and hypochlorite, depending on the pH (Equations (11)–(13)), thus oxidizing organics on the electrode interface and/or in the bulk of the solution (Equation (13)) [25,26].

$$2\,Cl^-_{(aq)} \rightarrow Cl^{\uparrow}_2 + 2e^-_{(aq)} \tag{10}$$

$$Cl^{\uparrow}_2 + H_2O \rightarrow HOCl_{(aq)} + Cl^-_{(aq)} + H^+_{(aq)} \tag{11}$$

$$HOCl_{(aq)} \leftrightharpoons H^+_{(aq)} + OCl^-_{(aq)} \tag{12}$$

$$Organics + OCl^-_{(aq)} \rightarrow intermediates \rightarrow CO_2 + Cl^-_{(aq)} + H_2O \;\; (alkaline\ medium) \tag{13}$$

For that reason, the use of low [NaCl] promotes an increase in SEC, due to the limited number of ions to be converted into reactive species and to the low conductivity of the solution, which provides an increase in the electric resistance of the system. For example, experiment 5 was conducted with all variables at their negative level, except $X_3$; in experiment 13, the coded values of $X_1$ and $X_2$ were negative and $X_3$ and $X_4$ were positive. Under these conditions, SEC values ranged from 10.06 to 6.44 kWh g$^{-1}$; that is, the increase in [NaCl] provided a twofold decrease in specific energy consumption.

Knowing the evolution of the concentration of Cl$^-$ ions is necessary to test the hypothesis: high values of current density improve the electro generation of active species. Table 6 presents the consumption of Cl$^-$ ions after 20 min of electrolysis for different process conditions, as previously established by the CCD matrix (Table 4).

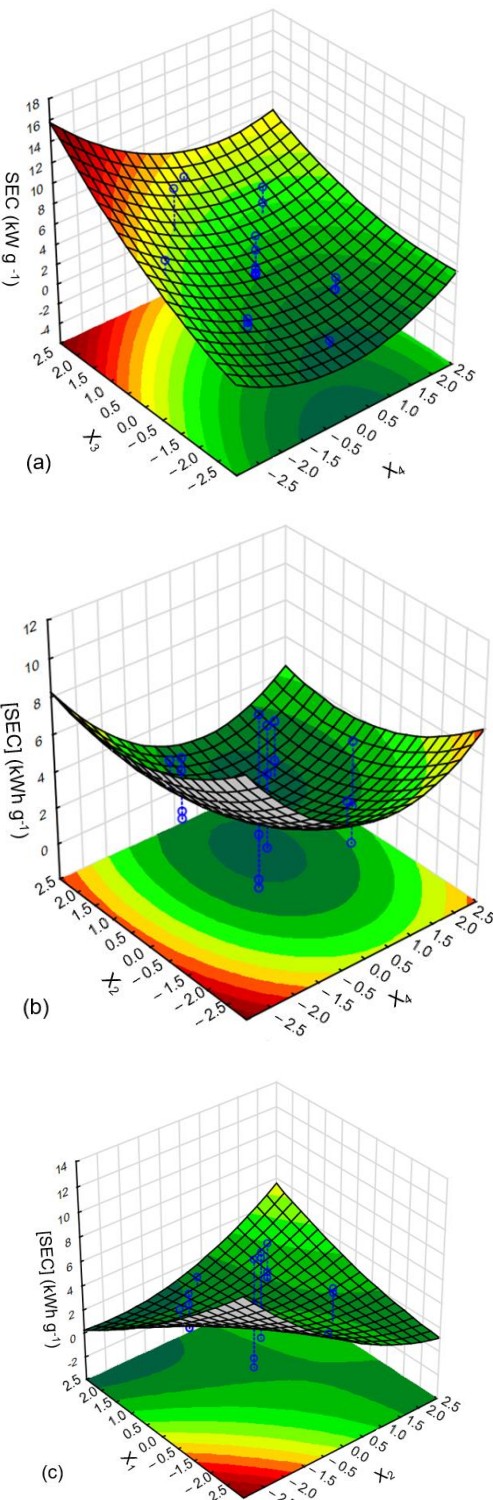

**Figure 2.** Three-dimensional surface plot of the specific energy consumption (*SEC*) as a function of the independent variables $X_1$ ([GES]$_0$), $X_2$ [LNG]$_0$, $X_3$ (*j*) (), and $X_4$([NaCl]$_0$) ). Each plot highlights the effects of two variables on the SEC with the other two variables fixed at their central point conditions: (a) $X_3$ (*j*) and $X_4$ ([NaCl]$_0$); (b) $X_2$ [LNG]$_0$ and $X_4$ ([NaCl]$_0$);c) $X_1$ ([GES]$_0$) and $X_2$ [LNG]$_0$.

**Table 6.** Consumption of chloride anions after 20 min of anodic oxidation for selected experiments.

| Run | $j$ (mA cm$^{-2}$) | $[NaCl]_0$ (mol L$^{-1}$) | Cl$^-$ Consumption (%) |
|---|---|---|---|
| 1 | 20 | 0.04 | 100 |
| 7 | 45 | 0.04 | 61.9 |
| 11 | 20 | 0.10 | 16.3 |
| 15 | 45 | 0.10 | 28.1 |
| 17 | 32.5 | 0.01 | 49.4 |
| 18 | 32.5 | 0.13 | 22.3 |
| 19 | 7.5 | 0.07 | 100 |
| 20 | 57.5 | 0.07 | 6.5 |
| 27 | 32.5 | 0.07 | 20.6 |
| 28 | 32.5 | 0.07 | 19.2 |

According to Table 6, the anodic oxidation of progestins provided a consumption of Cl$^-$ as a function of the initial concentration of NaCl and the applied current density. For example, runs 1 and 19 were performed with low values of $j$ = 20 and 7.5 mA cm$^{-2}$, respectively. In both experimental conditions, the electrolyte was applied at the lowest level of the empirical domain (0.04–0.07 mol L$^{-1}$). Consequently, runs 1 and 19 consumed 100% of the Cl$^-$ present in the solution to electrogenerated active species providing average progestin removals of 100% and 96.4%, respectively.

In contrast, run 20 was performed with a high level of $X_2$ and an intermediate level of $X_4$. In this condition, the maximum consumption of chlorine ions was 6.5%. Furthermore, the progestin removals achieved were 45.4% and 87.7% for LNG and GES, respectively. These results confirm the significant effect interaction among all studied variables and highlight that the use of high current densities does not always promote high generation of reactive species and consequently high removals of the target pollutants.

As can be seen in Figure 2b, at the central and constant levels of $[LNG]_0$ (1.0 mg L$^{-1}$) and [NaCl] (0.07 mol L$^{-1}$), the SEC tends to be a minimum. The interaction coefficient between the variables showed a significant effect due to the *p*-value < 0.001, which was also observed in Equation (9), where the interaction coefficient was $-0.7170X_2X_4$, indicating that to obtain a low SEC, it is necessary to operate the electrochemical system at low concentrations of both variables.

The effect of progestin concentrations on SEC can be observed in Figure 2c which has their levels set to central and constant levels. The use of low concentrations of LNG and GES (lower than $-1.5$ at the coded level) contributes to the increase in SEC. Nonetheless, the proportional increase of both progestins favors the process, and thus the specific energy consumption can be controlled. The negative interaction coefficient confirms this assumption, being a significant effect (*p*-value < 0.001).

### 3.1.2. Definition of Optimum Condition

Based on this set of evidence, it was possible to determine the optimal point of the experimental design (Table 7). However, for the concentrations of GES and NaCl used, the conditions suggested as optimal are outside the experimental domain, i.e., despite the practical possibility of these conditions, the result to be obtained will be outside the 95% statistical confidence interval, considered throughout the analysis.

**Table 7.** Values obtained as the optimum point, coded and actual, and central point.

| Variable | Optimum Point (coded) | Optimum Point (actual) | Central Point |
|---|---|---|---|
| $[GES]_0$ (mg L$^{-1}$) | 3.944 [a] | 2.97 | 1.0 |
| $[LNG]_0$ (mg L$^{-1}$) | 0.623 | 1.31 | 1.0 |
| $j$ (mA cm$^{-2}$) | $-1.432$ | 14.6 | 32.5 |
| $[NaCl]_0$ (mol L$^{-1}$) | 2.356 [a] | 0.14 | 0.07 |

[a] value outside the experimental domain.

Given that, the central point was chosen as the practical optimum point, corresponding to an SEC value of $2.2 \pm 0.2$ kWh g$^{-1}$ and removal efficiencies of $82.0 \pm 0.7\%$ for LNG and $75.0 \pm 0.5\%$ for GES (Table 4). The accuracy of the model was also verified through two experiments carried out with real pharmaceutical wastewater (RPW) under optimal practical conditions. As result, the anodic oxidation treatment of RPW samples under central point conditions ([NaCl]$_0$ = 0.07 mol L$^{-1}$, $j$ = 32.5 mA cm$^{-2}$, [LNG]$_0$ = [GES]$_0$ = 1.0 mg L$^{-1}$) resulted in an SEC of $2.32 \pm 0.03$ kWh g$^{-1}$ and removal efficiencies of $83.1 \pm 0.9\%$ for LNG and $75.1 \pm 2.8\%$ for GES. Furthermore, considering the average price of USD 0.13 per kWh in Brazil [20], the estimated cost for treating wastewater containing LNG and GES is equivalent to about USD 0.26 m$^{-3}$, under the conditions of the central point (Table 7). The motivation is the future implementation of anodic oxidation in wastewater treatment plants. In addition, the application of electrochemical oxidation technologies in other point sources, such as in the treatment of hospital effluents [51], pharmaceutical effluents [52], and agro-industrial effluents [53], can be seen as an opportunity to reduce the impacts caused by organic contaminants of emerging concern.

### 3.2. Estrogenic Activity Removal

The investigation of estrogenic activity was conducted by the YES assay as described in Section 2.5. E2 was used as a standard for obtaining a dose-response curve (Figure S3). Consequently, the sample treated under central point conditions (run 28, [NaCl]$_0$ = 0.07 mol L$^{-1}$, $j$ = 32.5 mA cm$^{-2}$, [LNG]$_0$ = [GES]$_0$ = 1.0 mg L$^{-1}$) was used in the assay, which was also represented by the dose-response curve (Figure S4).

Figure 3 presents the removal of estrogenic activity observed for the LNG + GES solution before the AO process which reached a value of 924 E2-EQ ng L$^{-1}$. This value was considered high compared with E2 which had an EC50 of 49.5 ng L$^{-1}$. In addition, Figure 3 shows the efficiency of the AO process in reducing the initial estrogenic activity of the sample by 74.6%, achieving an average of 234 E2-EQ ng L$^{-1}$. This result suggests that the degradation of progestin molecules did not form organic by-products with functional groups associated with the estrogenic activity of the parent compound, nor phenolic by-products after the anodic oxidation process.

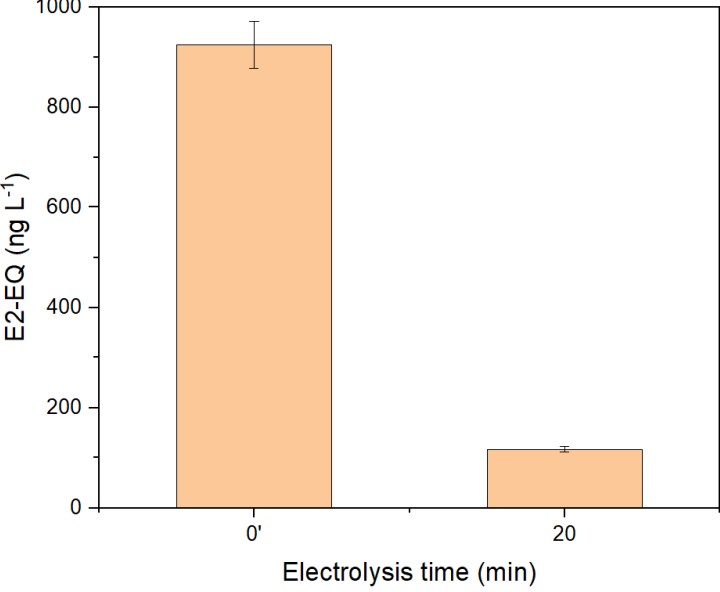

**Figure 3.** Results of estrogenic activity (E2-EQ) of LNG-GES solutions before and after 20 min of anodic oxidation mediated by chlorine active species ([NaCl]$_0$ = 0.07 mol L$^{-1}$, $j$ = 32.5 mA cm$^{-2}$, [LNG]$_0$ = [GES]$_0$ = 1.0 mg L$^{-1}$). Measurements performed in duplicate.

Saggioro et al. [54] investigated the use of UV/chlorine to degrade bisphenol-A (BPA), a known endocrine disruptor that also has estrogenic activity. First, untreated solutions were evaluated ($[BPA]_0$ = 100 μg $L^{-1}$ and $[Cl^-]_0$ = 2 mg $L^{-1}$) showing ~280 μg $L^{-1}$ E2-EQ. After 2 and 5 min of UV irradiation, both reached estrogenic activities below 26.0 ±12.0 ng·$L^{-1}$ EQ-E2. Furthermore, Saggioro et al. [54] concluded that the samples evaluated did not produce cytotoxic by-products for *S. cerevisiae* cells.

Cunha et al. [55] evaluated the feasibility of the electrochemical filter with a carbon nanotube to remove E2 ethinylestradiol (EE2) and estrogenic activity. The authors applied an electrochemical system consisting of a titanium cathode and a carbon nanotube anode filter. Sodium sulfate (10 mmol $L^{-1}$) was used as the supporting electrolyte and a voltage rate (0–2.5 V) was applied during 300 min of electrolysis. Under these conditions, the YES assays allowed verification that the process efficiently reduced the estrogenic activity of the target hormones (2.467 ± 0.0012 ng·$L^{-1}$ E2-EQ). However, even under ideal conditions (2.5 V), mineralization was not achieved.

### 3.3. Toxicity Effect on D. similis

Samples A–E (Section 2.6) were applied to the assay as controls to investigate the effect of the electrogenerated species before and after the anodic oxidation process and to verify whether the effect on acute ecotoxicity on *D. similis* is a result of hormones, chlorinated species, or a combination of both (Figure 4). From this, the three mentioned conditions showed similar UT values (~3.0 UT 100/CE50%). This result suggests that the presence of NaCl in the mixture of LNG and GES does not interfere with the toxic effect attributed to the hormones before electrolysis.

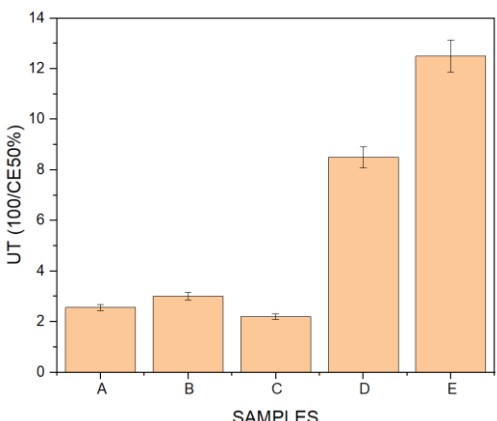

**Figure 4.** Results of acute toxicity tests (in toxicity units, TU= 100/EC50%) to *D. similis* before and after anodic oxidation for samples (**A**) mixture of progestogens (LNG + GES) without electrochemical processing and in the absence of NaCl; (**B**) mixture of progestogens (LNG + GES) without electrochemical processing in the presence of NaCl; (**C**) NaCl solution without electrochemical processing; (**D**) NaCl solution after electrochemical processing (20 min); and (**E**) mixture of progestogens (LNG + GES) after electrochemical processing (20 min) in the presence of NaCl ($[LNG]_0$ = $[GES]_0$ = 1.0 mg $L^{-1}$; *j* = 32.5 mA $cm^{-2}$; $[NaCl]_0$ = 0.07 mol $L^{-1}$). Values correspond to an average of eight replicates.

After the oxidation process using the DSA-$Cl_2$ system, the saline solution (sample D) showed an increase in UT of about three times, when compared to that observed for the untreated solution (sample C). Finally, sample E indicated an even greater increase of about four times that observed for sample B. In the latter case, the relevance of the process can be verified, since after the treatment there was the formation of chlorinated species, which may have contributed to the intensification of the acute effect to the organisms evaluated (Figure 4).

Lin et al. [34] applied bioassays with *D. magna* to monitor the degradation of orange II by the electro-Fenton (EF) process. The authors observed that the sample obtained by the

electrochemical process after 5 min resulted in 100% immobilization of the test organisms. However, after 60 min of the treatment process, no immobilization was observed, thus confirming the feasibility of EF in removing acute toxicity.

## 4. Conclusions

The anodic oxidation process proved to be effective in decomposing progestins in a synthetic mixture and real pharmaceutical wastewater (RPW), with removals $\geq 70\%$ in less than 20 min of electrolysis. The statistical analysis of the data allowed for identifying the main effects of first-order, second-order, and interaction of the independent variables ($[LNG]_0$, $[GES]_0$, $j$, and $[NaCl]_0$). The statistical model obtained allowed for determining the relative importance of these variables on the response specific energy consumption (LNG concentration > NaCl concentration > current density > GES concentration). The model presented a correlation coefficient of 89.5%, indicating a good fit to the data with significant F and *p*-values, allowing estimation of the optimal point which, for practical purposes, was assumed as the central point ($[LNG]_0 = 1.05 \pm 0.03$ mg L$^{-1}$ and $[GES]_0 = 1.10 \pm 0.17$ mg L$^{-1}$; current density, 32.5 mA cm$^{-2}$; and $[NaCl]_0 = 0.07$ mol L$^{-1}$). These conditions correspond to an SEC value of $2.2 \pm 0.2$ kWh g$^{-1}$ and removal efficiencies of $82.0 \pm 0.7\%$ for LNG and $75.0 \pm 0.5\%$ for GES. The model accuracy was also verified under central point conditions for RPW, resulting in an SEC of $2.32 \pm 0.03$ kWh g$^{-1}$ and removal efficiencies of $83.1 \pm 0.9\%$ for LNG and $75.1 \pm 2.8\%$ for GES. Regarding estrogenic activity, the treatment under central point conditions allowed the reduction of estrogenicity by 74.6% after 20 min of anodic oxidation. Finally, the evolution of acute toxicity, evaluated with *D. similis*, proved to be strongly dependent on the presence of chlorinated species; however, prolonged anodic oxidation treatment can reduce residual ecotoxicity, as reported in the literature.

**Supplementary Materials:** The following supporting information can be downloaded at: https://www.mdpi.com/article/10.3390/w15010061/s1, Figure S1—(I) Electrochemical apparatus used for the degradation of progestogens via anodic oxidation: (1) reservoir; (2) peristaltic pump; (3) electrochemical cell; (4) power supply; (5) multimeter. (II) Schematic representation of the electrochemical apparatus. (III) Disassembled electrochemical cell; Figure S2—Pareto chart of standardized effects; Figure S3—Dose-response curve for the positive control E2 (acetonitrile: water of 55:45 *v/v*. LOD = 2.3 ng L$^{-1}$ and LOQ = 7.0 ng L$^{-1}$); Figure S4—Dose-response curve for experiment 28 before (a) and after (b) the anodic oxidation process ($[NaCl]_0 = 0.07$ mol L$^{-1}$, $j = 32.5$ mA cm$^{-2}$, $[LNG]_0 = [GES]_0 = 1.0$ mg L$^{-1}$); Table S1——Physical and chemical properties of progestins under study; Table S2—Validation parameters of the LNG and GES calibration curves obtained by UFLC analysis. LOD: limit of detection; LOQ: limit of quantification; RSD: relative standard deviation; CI: confidence interval.

**Author Contributions:** Conceptualization, J.M.S.d.J.; methodology, J.M.S.d.J., A.d.S.A. and F.K.T.; software, J.M.S.d.J. and M.E.T.; validation, J.M.S.d.J.; formal analysis, J.M.S.d.J. and M.E.T.; investigation, J.M.S.d.J.; resources, S.I.B., D.M.B. and A.C.S.C.T.; data curation, J.M.S.d.J., A.d.S.A., F.K.T., and A.C.S.C.T.; writing—original draft preparation, J.M.S.d.J.; writing— review and editing, J.M.S.d.J. and A.C.S.C.T.; visualization, J.M.S.d.J.; supervision, M.E.T. and A.C.S.C.T.; project administration, A.C.S.C.T.; funding acquisition, A.C.S.C.T. All authors have read and agreed to the published version of the manuscript.

**Funding:** This research was funded in part by the Coordenação de Aperfeiçoamento de Pessoal de Nível Superior, Brasil (CAPES), Finance Code 001. The authors thank the support of the São Paulo Research Foundation (FAPESP) [grant number 2019/24158–9] and the National Council for Scientific and Technological Development—Brazil (CNPq) [grant number 311230/2020–2].

**Institutional Review Board Statement:** Not applicable.

**Informed Consent Statement:** Not applicable.

**Data Availability Statement:** Not applicable.

**Conflicts of Interest:** The authors declare no conflict of interest. The funders had no role in the design of the study; in the collection, analyses, or interpretation of data; in the writing of the manuscript; or in the decision to publish the results.

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
