# Peer review of "Experimental Design and Bioassays as Tools to Investigate the Impact of Anodic Oxidation on Progestins Degradation"

_water, doi:10.3390/w15010061_

Round 1

Reviewer 1 Report

The article as a whole is well written, well referenced. Although, clear indications on the application of the presented research are lacking.

The mathematical and statistical models enrich the work, but it lacks specificity when it comes to showing the results.

It is complicated as the article is structured to clearly understand the purpose of the research presented.

The method is sufficiently described, but the results and, to a greater extent, the conclusions appear confusing.

Greater coherence between the objectives and the conclusions presented is required.

On page 13 of 19 "error" appears several times, problems with references.

Therefore, the article has a limited degree of novelty and the relationship between objectives, results and conclusions should be reviewed and defended in more detail.

Author Response

Dear reviewer, 

Thanks for your revision. Check the responses to your questions in the attachment.

Regards,

Reviewer 2 Report

1 Ln263-264, “a quadratic equation (Eq.10)” may be Eq.9.

2 Ln267-273, the author should explain the quadratic equation (Eq.9). which one or interaction coefficients is the significant parameter (s)?

3 Ln318-321, some Eqs. and Reference are not displayed and showed errors.

4 Ln349-350, what was reason for that high current density leading to low removal efficiency?

5 Ln385-388, the first reported results that by-products of electrooxidation did not enhance the initial estrogenic activity, what is the essential reason?

6 Ln412-416, the sample information should be set in the M&M.

Author Response

(The authors gave the same response as above.)

Round 2

Reviewer 2 Report

Ln278, "3 Results" should be "Results and Discussion"

Ln496, "3.6 Toxicity effect on D. similis" should be "3.3 Toxicity effect on D. similis"

Author Response

(The authors gave the same response as above.)
